# Transcription Factors in Plant Stress Responses: Challenges and Potential for Sugarcane Improvement

**DOI:** 10.3390/plants9040491

**Published:** 2020-04-10

**Authors:** Talha Javed, Rubab Shabbir, Ahmad Ali, Irfan Afzal, Uroosa Zaheer, San-Ji Gao

**Affiliations:** 1National Engineering Research Center for Sugarcane, Fujian Agriculture and Forestry University, Fuzhou 350002, China; talhajaved54321@gmail.com (T.J.); rubabshabbir28@gmail.com (R.S.); ahmad03348454473@yahoo.com (A.A.); uroosazaheer58@gmail.com (U.Z.); 2Seed Physiology Lab., Department of Agronomy, University of Agriculture, Faisalabad-38040, Pakistan; iafzal@uaf.edu.pk

**Keywords:** *Saccharum* spp., transcription factor, gene regulation, genetic improvement, biotic and abiotic stresses

## Abstract

Increasing vulnerability of crops to a wide range of abiotic and biotic stresses can have a marked influence on the growth and yield of major crops, especially sugarcane (*Saccharum* spp.). In response to various stresses, plants have evolved a variety of complex defense systems of signal perception and transduction networks. Transcription factors (TFs) that are activated by different pathways of signal transduction and can directly or indirectly combine with *cis*-acting elements to modulate the transcription efficiency of target genes, which play key regulators for crop genetic improvement. Over the past decade, significant progresses have been made in deciphering the role of plant TFs as key regulators of environmental responses in particular important cereal crops; however, a limited amount of studies have focused on sugarcane. This review summarizes the potential functions of major TF families, such as WRKY, NAC, MYB and AP2/ERF, in regulating gene expression in the response of plants to abiotic and biotic stresses, which provides important clues for the engineering of stress-tolerant cultivars in sugarcane.

## 1. Introduction

During the life cycle of plant species, abiotic (temperature, waterlogging, drought, oxidative, salinity, ultraviolet) and biotic (viruses, bacteria, fungi, insects etc.) stresses are major threats to global crop productivity [1]. To cope with various stresses, plants have evolved complex rapid responses, but crop efficiency is still severely hampered. The prevailing alarming climate change scenario and future changing weather provide challenges for the researchers to better understanding plant responses such as the activation of different signal cascades, followed by signal transduction, and the respective gene expression in response to single or combined biotic or abiotic stresses, ultimately enhancing crop yield [2]. Sugarcane (*Saccharum* spp.) plays a crucial role in sugar and biofuel production, accounting for 80% of sugar production in the world. Genetic improvement of modern sugarcane cultivars hits a bottleneck through conventional hybrid breeding as a result of sugarcane complex genome, heterogenous and polyploid-aneuploid nature [3]. 

Under unfavorable environmental conditions, various stress-response mechanisms have been evolved by plants at different layers, such as cellular signal perception and transduction, inducing expression of specific sub-sets of defense genes, and therefore activating the overall defense reaction, ultimately contributing to phenotype [4]. Transcription factors (TFs) are important regulators for the control of gene expression in all living organisms and play crucial roles in plant development, cell cycling, cell signaling and stress response [5]. TFs modulate gene expression through binding with their distal and local *cis*-elements of target gene, which might be influenced by genomic features, DNA structure and TF interactions [6]. In plants, TFs are encoded by approximately 10% of genes at different stages/points to regulate specific signaling mediated function [5]. Various TFs databases specific for various crops are available, such as the plant transcription factor database and grass TF database. Major TF families, such as WRKY, MYB, NAC and AP2/ERF, are crucial regulators of various genes related to different stresses, which contribute to the ideal choice for genetic engineering in order to enhance resistance of plants against different stress stimuli [7]. According to the plant TF database, numerous TFs have been reported in *Oryza sativa* (2389), *Hordeum vulgare* (2620), *Triticum aestivum* (3437) and *Saccharum* spp. (672). It is noteworthy that, to date, 39 WRKY, 44 NAC, 38 MYB and 73 AP2/ERF gene families of TFs in sugarcane have been identified.

In last decade, numerous TFs involved in response to abiotic and biotic stress in different crops including sugarcane were investigated using modern molecular tools, such as functional genomics, transcriptomics and proteomics [8]. This review provides recently updated regulatory functions of major TFs families in response to stresses to understand the stress-responsive mechanisms in various crops, including sugarcane.

## 2. WRKY TFs in Response to Stresses

### 2.1. Classification and Diversification of WRKY Genes Family

WRKY being as one of the most characterized class of plant TFs, regulate diverse plants developmental, physiological and metabolic processes [9]. The first WRKY TF was identified in *Ipomoea batatas* in 1990s [10], but WRKY genes were considered plant specific TFs at prehistoric times [2]. A myriad of studies documented that other eukaryotic organisms, such fungi, amoebae and diplomonads, also have WRKY proteins in their genetic make-up. Ancient gene transfer events were considered as the origin of these non-plant WRKYs based on distribution pattern analysis [11]. Numerous WRKY TFs have been experimentally revealed in various plant species, including (*Arabidopsis thaliana*) [12], sugarcane (*Saccharum spontaneum*) [13], barley (*Hordeum vulgare*) [14], rice (*Oryza sativa*) [15], physic nut (*Jatropha curcas*) [16], maize (*Zea mays*) [17], wheat (*Triticum aestivum*) [18], cotton (*Gossypium hirsutum*) [19] and so on (Table 1).

Based on the genomic characterization of *Arabidopsis*, the most authenticated and widely accepted classification system stipulates that plant specific WRKY TFs are denoted by 60 amino acids in their highly conserved DNA binding region (called as WRKY domain) [20]. Furthermore, WRKY domains compromised of a highly conserved motif WRKYGQK at N terminus that provides a protein-protein interaction interface and a zinc finger region either C-X_4-5_-C-X_22-23_-H-X-H or C-X_7_-C-X_23_-H-X-C at the C-terminus having affinity towards DNA binding [21]. WRKY proteins, based on WRKY domains present and zinc finger region, have been classified into three major groups (Group I, II and III) [22]. Group I have two WRKY domains with a zinc finger motif (C-X_4-5_-C-X_22-23_-H-X-H) adjacent to atypical. Group II and III WRKYs have one WRKY domain. WRKYs in group I and II have the same type of finger motif, whereas group III WRKYs consists of a C-X_7_-C-X_23_-H-X-C motif at the C-terminus. WRKYs in Group II can be further categorized into five sub-groups (IIa, IIb, IIc, IId and IIe), while WRKYs in Group III can be categorized into two groups, IIIa (C-X_7_-C-X_23_-H-X-C) and IIIb (C-X_7_-C-Xn-H-X-C, n ≥ 24), based on the zinc-finger motif structure [21]. However, an additional WRKYs Group IV was proposed in *S. spontaneum*, as evidenced by these genes containing an incomplete domain (only the WRKYGQK motif was identified), which indicated they may have lost their function as WRKYs [13]. Taking together, the biological functions of all WRKYs groups have been linked with their ability to bind specifically in the promoter region of target genes to the W-box, i.e., (C/T) TGAC (C/T) and modulate these gene expressions [23].

### 2.2. Function and Expression Pattern of WRKYs under Abiotic Stresses

High or low temperature stress has significant influence on a myriad of processes including physiological, enzymatic, metabolic, growth and developmental in plant system, hence leading to low yield with poor quality of produce [9]. More specifically, decreased enzymatic activity due to denatured entire protein structure under temperature extremes lead to the halt of entire pathway, thereby resulting in wilting or senescence of the plant [24]. Stimulation of heat-induced signal transduction pathway by the WRKYs group I proteins (such as *AtWRKY25*, *AtWRKY26* and *AtWRKY33*) contributed to significantly higher thermo-tolerance. While double (*wrky25/26*) and triple (*wrky25/26/33*) mutants depict poor seed germination, decreased membrane integrity and increased susceptibility to temperature stress [25]. The constitutive synergistic expression of *TaWRKY1* and *TaWRKY33* in *Arabidopsis* [26] and *TaWRKY70* [27] in *T. aestivum* exhibits enhanced tolerance to heat stress. Furthermore, overexpression of *TaWRKY008, TaWRKY122* and *TaWRKY45* in *T. aestivum* also enhanced tolerance to extreme temperature stress [28]. Overexpression of *VaWRKY12* that was localized in the nucleus inhibited the cellular damage in *Arabidopsis* and grapevine after cold stress [29], while overexpression of *ZmWRKY106* in *Zea mays* showed higher activities of antioxidants under heat stress and improved vigor of transgenic plants [30].

Waterlogging is one of the serious threats to agriculture, as it affects the aeration ability of roots and alters the physical and chemical nature of soil, thus triggering the adverse effects on productivity of crop species worldwide [31]. Overexpression of *WRKY22* in *Arabidopsis*, by targeting genes encoding a toll/interleukin-1 receptor (TIR) domain–containing protein, lead to cope the threat associated with excess soil moisture [32]. A sunflower TF *HaWRKY76* was upregulated in *Arabidopsis* transgenic plants in response to excess soil moisture, imparting enhanced resistance through the repression of fermentation pathways that ensue to carbohydrates reservation [33]. In contrast, *PlWRKY70*, a *Paeonia lactiflora* TF, was considerably suppressed under waterlogging stress [34]. 

Drought, due to reduced potential water content in plants, can directly limit the normal functioning of crop plants by averting the enzymatic activities, accumulation of soluble matters, formation of ROS (reactive oxygen species) multiplexes and restricted metabolic pathways, thus making the agricultural efficiency at risk [35]. WRKY TFs has been frequently identified and reported to play optimistic role in regulating defense genes under water deficient environments. For example, *AtWRKY28* in *Arabidopsis*, positively regulated in response to drought stress [36], whereas *WRKY46/54/70* acted as negative regulators under drought stress [37]. In sugarcane, a WRKY-IIc TF gene *ScWRKY3* played positive regulation by sodium chloride (NaCl), polyethylene glycol (PEG), and abscisic acid (ABA), but it was suppressed by salicylic acid (SA) and methyl jasmonate (MeJA) [38]. In other crops, overexpression of *TaWRKY93* [39], *ZmWRKY40* [17], *TaWRKY2* [40], *GhWRKY91* [41] and *GmWRKY54* [42] conferred drought tolerance through ABA, Ca^2+^-mediated signal transduction pathways and osmotic adjustments. 

Ultraviolet (UV) light plays a crucial role from germination to the growth and development in plants [43]. However, UV type B light can restrict the normal enzymatic functioning and also cause significant damage to DNA bases. In rice, overexpressing *OsWRKY89*, a group III TF, through direct or indirect activation of SA mediated signal transduction pathway, not only intensified the plants resistance against UV irradiation but also imparted tolerance counter to insect pests and fungal infection [44]. Oxidative stress mediated through jasmonate (JA) signaling pathway is among one of the most detrimental stresses [45], while production and prompt scavenging of ROS are responsible for regulation of ROS-mediated signaling in plants [46]. Overexpression of *WRKY4*, *WRKY5*, *WRKY11* and *WRKY46* were observed during the exposure of oxidative stress in *Salvia officinalis* and play a pivotal role in the signaling mechanism during plant responses [45].

### 2.3. Function and Expression Pattern of WRKY Genes under Biotic Stresses 

During biotic stress, plants undergo the partial or complete modulation of multiple signal transduction pathways such as plant hormones, which subsequently resulted in the induction of several related transcriptional genes, thus resulting in positive response to stressful environment [4]. Plant WRKYs have been involved in the microbe-associated molecular pattern-triggered immunity (PAMP-triggered immunity), effector-triggered immunity (ETI), or system acquired resistance (SAR) [9]. For example, *CsWRKY50* plays crucial role against infection stress of *Pseudopernospora cubensis* in *Cucumis sativus* [47]. *VvWRKY1*, JA pathway-related gene, positively regulates the stress tolerance response to downy mildew in *Vitis vinifera* [48]. In addition, another JA-regulated gene, *CaWRKY27*, confers the resistance against *Ralstonia solanacearum* infection in tobacco (*Nicotiana tabacum*) [49]. *GhWRKY44*, a significant TF in cotton-pathogen interaction, it provides an enhanced tolerance in tobacco against bacterial and fungal pathogens. The plants exhibited a lower level of ROS accumulation in experimental units having overexpression of *GhWRKY44* through SA and JA signal transduction pathways of disease resistance [50]. A significant increase in resistance by *TaWRKY70* transcription factor, through SA and ET mediated signal transduction pathways, was noted in plants when exposed to pathogen (*Puccinia striiformis* sp. *tritici*) infection [27]. A versatile role in response to pathogen stimuli was indicated by *WsWRKY1* [51], *AcWRKYs* [52] and *GmWRKY31* [53] TFs in different crops. The sugarcane *ScWRKY3* was stably expressed in the smut-resistant cultivar, whereas it was suppressed in the smut-susceptible cultivar at early stages (0–72 h) of infection with the smut pathogen (*Sporisorium scitamineum*). Meanwhile, this gene was proposed to act as a negative regulator under the pathogen infection either *Fusarium solani* var. *coeruleum* or *Ralstonia solanacearum* in *N. benthamiana* [38]. More recently, most of *WRKY33* alleles were significantly upregulated in sugarcane against the *Xanthomonas albilineans* attacks [54]. Collectively, WRKY TFs through self-regulation or hormones mediated signal transduction pathways proved to be effective for alleviating the infection stress caused by biotic or abiotic agents.

## 3. NAC TFs in Response to Stresses

### 3.1. Classification and Diversification of NAC Gene Family

NAC (NAM: no apical meristem, ATAF, CUC: cup-shaped cotyledon) is one of the most important and largest family of plant-specific stress-responsive TFs [55]. *NAM* from *Petunia hybrida* [56] and *Ataf1/2* from *Arabidopsis* [57] were the first reported NAC proteins. Numerous NAC genes have been identified in sugarcane and other important crops (Table 1). NAC TFs are characterized by diverse C terminal that contains a variable transcriptional regulatory region (TR), while N-terminal equipped with 150–160 amino acids and also harbor NAC domain for DNA binding [58,59]. The highly conserved NAC domain can be further classified into five different sub-domains (A–E). Furthermore, nuclear localization, formation of homo/heterodimers and DNA binding have been associated with the functions of conserved NAC domain, while TR region either as activator or repressor was proven to be associated with transcription regulation [59]. Based on structure, NAC TFs can be grouped into two sub classes, typical and atypical NAC TFs. Typical NAC TFs are characterized by the presence of NAC domain at N terminal and a divergent C-region [59], while in atypical NAC TFs, the C-terminal regions contain additional motifs/domains or the C-terminus can be absent [60]. NTLs (NAC with transmembrane motif1-like), atypical NAC TFs, are characterized by the occurrence of transmembrane (TM) motif in C-region [61]. Notably, TM motif is thought to be involved in anchoring to plasma membrane, where they could impart their function by releasing through proteolysis [62,63].

### 3.2. Function and Expression Pattern of NAC TFs under Abiotic Stresses 

Crop plants employ various molecular mechanisms such as chaperon signaling, ROS scavenging, various compatible solutes accumulation, transcriptional activity, antioxidant production and induction of mitogen-activated protein kinase (MAPK) and calcium-dependent protein kinase (CDPK) cascades against heat stress [64,65]. NAC TFs were observed to be involved in molecular regulations during temperature fluctuations. For example, overexpression of *NTL1* and *NTL11* genes were observed after heat stress, while *NTL4* and *NTL7* were exhibited specifically by low temperature in *Arabidopsis* [62]. In sugarcane, *SsNAC23* expression was induce in response to low temperature stress [66]. *ANAC019*, member of NAC TFs, has been observed to enhance the ability of *Arabidopsis* to cold tolerance by inducing the expression of cold responsive marker gene *COR47* [55]. Transformation of *Arabidopsis* with *GmNAC20* of *Glycine max* exhibits an increased tolerance against chilling and salinity stress by the activation of dehydration responsive element binding/C-repeat binding factor-cold responsive (DREB/CBF-COR) pathway [67]. Various members of NAC play a significant role against temperature stress in different crop species as evidenced by overexpression of *OsNAP* exhibit improved resistance in rice against cold stress at the vegetative stage through ABA-mediated signal transduction pathway [68]. Notably, interaction of *LlNAC2* with *LlZHFD4* and *LlDREB1* from tiger lily enhanced the tolerance in *Arabidopsis* against cold, salinity and drought stresses by maintaining its membrane integrity [69]. *CaNAC064* was found a positive modulator of plant resistance to cold stress by interacting with haplo-proteins that were induced by low temperature [70].

Aquaporins expression, stability of cell membrane, accumulation of osmo-protectants and scavenging of ROS are modulated in plants when exposed to drought stress [71]. NAC TFs have been reported to regulate drought stress in *Arabidopsis* as overexpressing *ANAC019*, *ANAC055* and *ANAC072* genes bind specifically to drought responsive *cis*-element [72]. Overexpression of *OsNAC10* (root-specific) exhibited to improve root dynamics and to enhance drought tolerance, which ultimately leading towards sustainable plants physio-morphological and yield related attributes in rice under field conditions [73]. Another root specific *OsNAC6* TF was proven to be effective for enhanced drought tolerance and root development in rice [74]. In transgenic rice, overexpression of *EcNAC67* upon drought stress displayed higher seedling vigor, lower spikelet sterility and enhanced tolerance to drought stress [75]. Overexpression of *TaNAC47* and *ONAC066* enhanced drought tolerance in wheat [76] and rice [77], respectively. Moreover, *GhNAC2* [78] showed the potential to enhance root growth and tolerance against limited water supply in transgenic cotton and *Arabidopsis*. In *Pyrus betulifolia*, overexpression of *PbeNAC1* imparted resistance against chilling and drought stress by interacting with *PbeDREBs* [79]. Few overexpressing NAC genes such as *NAC016* in *Arabidopsis* [80] and *ShNAC1* in tomato [81] were also found to be negative regulators against drought stress. 

Balazadeh et al. [82] demonstrated the disruptive role of *ANAC092* for higher rate of seed germination, salinity inducing lower chlorophyll loss at later stages, and the delay of leaf and flower senescence in *Arabidopsis*. Significantly enhanced tolerance to osmotic stress in rice was attributed by *OsSRO1c* gene. This gene imparts stomatal closure and hydrogen peroxide (H_2_O_2_) accumulation by interacting with several stress responsive functional and regulatory proteins [83]. In soybean, lateral root development, alleviation of salt and freezing stress was resulted from the overexpression of *GmNAC20*. Stress alleviation by *GmNAC20* was regulated by the activation of the DREB/CBF–COR pathway. Overexpressing *GmNAC11* only led to decreased sensitivity towards salinity [67]. A nucleus localized TF *TaNAC67* conferred significantly enhanced multi-abiotic stresses (drought, salt, freezing) tolerance in transgenic *Arabidopsis* [84]. Overexpression of *TaNAC29* in transgenic *Arabidopsis* plants, grown under greenhouse conditions, showed decreased sensitivity towards drought and salinity stresses with increased activity of antioxidants and lower accumulation of harmful malondialdehyde (MDA) and H_2_O_2_ contents [85]. Enhanced tolerance to salinity and osmotic stresses was markedly confirmed by overexpression of *ThNAC7* in *Tamarix hispida* seedlings [86].

### 3.3. Function and Expression Pattern of NAC TFs under Biotic Stresses 

Many studies showed that a number of NAC TFs play dual roles in plant immunity against various pathogens through the hypersensitive responses and ETI [87]. A wheat TF *TaNAC8* has a positive role to protect plants against the stripe rust pathogen infection [88]. *ZmNAC41* and *ZmNAC100* genes were induced by JA and SA pathways, respectively, to ensure tolerance in maize against *Colletotrichum graminicola* [89]. In contrast, *TaNAC30* [90] and *TaNAC2* [91] negatively regulated the defense mechanism against pathogen stress. Virus induced gene silencing (VIGS) analysis in tomato depicted the positive role of *SlNAC61* against infection stress caused by the Tomato yellow leaf curl virus (TYLCV) [92]. In rice, *ONAC122* and *ONAC131* TFs have imperative roles in diseases tolerance responses through the regulated expression of defense- and signaling-related genes such as *OsLOX*, *OsPR1a*, *OsWRKY45* and *OsNH1* [93]. In tomato, two homologous NAC TFs (*JA2* and *JA2L*) differentially regulated stomatal closure and reopening under pathogen attack through. Namely, *JA2* played a positive role in stomatal closure by regulating the expression of an ABA biosynthetic gene, whereas *JA2L* served as a positive role in JA/COR (JA/coronatine)-mediated stomatal reopening by regulating the expression of genes involved in the JA metabolism [94]. Interestingly, hypersensitive cell death in response to bacterial pathogens was promoted by the overexpression of *NAC4* in *Arabidopsis* [95]. The *LsNAC069*-silenced lettuce lines showed non-significant alteration in susceptibility to *Bremia Lactucae*, but increased resistance to *Pseudomonas cichorii* bacteria [96]. The characterization of TFs in sugarcane is still limited. Overall, NAC TFs through several signal mediated cascades play important role in protecting plants against different abiotic and biotic stresses. 

## 4. MYB TFs in Response to Stresses

### 4.1. Classification and Diversification of MYB Gene Family

The MYB family is a major and functionally diverse protein class of eukaryotes. The proteins of this family as TFs are majorly associated with protein-protein interaction, DNA binding and regulatory activity management of proteins [97]. Several MYB proteins have also been identified in regulating various cellular processes such as the stress responses, cell morphogenesis and cell cycle of different crop species [98]. The MYB gene “*colored1* (*c1*)” was identified in *Zea mays,* which encodes a MYB-protein domain involved in anthocyanin biosynthesis in aleurone layer of maize seed [99]. 

Based on the number of repeats (varying from 1–4) in sequence, MYB family proteins have been classified in four groups, i.e., 1R-MYB (one repeat), R2R3-MYB (two repeats), 3R-MYB (three repeats) and 4R-MYB (four repeats) [100]. Each repeat forms three α-helices containing approximately 50–53 amino-acids with second and third helices forming HTH fold (helix-turn-helix). HTH fold consists of three tryptophan (amino acids) spaced equally apart forming a hydrophobic core [101]. The R2R3-MYB subfamily has also been classified in 30–38 groups on the basis of diverse domains of N- and C-terminal [102]. The presence and role of MYB TFs have thoroughly been investigated in various plant species including sugarcane, thus making them key factors for regulating response to biotic and abiotic stresses (Table 1). 

### 4.2. Function and Expression Pattern of MYB TFs under Abiotic Stresses

Plants regulate the stress related genes and signaling networks to overcome abiotic stresses [2]. Many of MYB TFs have been observed to play a significant role in regulating heat stress of various crops. Like in tomato, overexpression of *LeAN2* improved the resistance of plants against heat stress through a high level of antioxidants activity (non-enzymatic) and a low level of ROS accumulation [103]. In rice, overexpressing *OsMYB1* gene can increase the tolerance of plants to both heat and salinity stresses [104]. While in maize, *OsMYB55* proved to be effective in enhancing tolerance against high temperature and drought stress with improved growth and development of seedlings [105]. In wheat, six MYB genes related to heat stress were identified out of which *TaMYB80* was effective for resistance against heat and drought stresses in transgenic *Arabidopsis* [106]. Similarly, *PbrMYB5* from *Pyrus betulaefolia* imparted tolerance against cold stress by modulating the ascorbic acid biosynthesis in tobacco [107].

MYB TFs have been observed to be involved in ABA signaling pathways in response to drought stress [108] in various crops particularly in *Arabidopsis.* Approximately 51% MYB proteins were upregulated (such as *AtMYB2/74/102*) and 41% were downregulated as a result of drought stress [109]. Overexpression of *AtMYB44* (a member of R2R3-MYB TF subfamily) has been observed to enhance the tolerance against drought/salt-stress in soybean [110]. Similarly, *BcMYB1*, another member of R2R3-MYB TF subfamily, was induced strongly during water deficiency, high polyethylene glycol contents and salinity stress in *Boea crassifolia*. However, only marginal induction of *BcMYB1* was observed in the case of low temperatures [111]. 

In *Arabidopsis*, overexpression of *AtMYB37* could increase seed yield and tolerance of transgenic plants against drought stresses [112]. In sugarcane, an MYB TF gene *PScMYBAS1* promoter was involved in positive regulation under abiotic stresses (dehydration, salt, cold and wounding) and hormone (SA and MeJA) treatments [113]. Recently, a study reported that the overexpression of *ScMYBAS1-3* was associated with drought resistance and biomass accumulation in rice transgenic lines [114]. Another observation revealed that a sugarcane *ScMYB2* was participated in an ABA-mediated leaf senescence signaling pathway and acted as a positive regulator in respond to PEG-mediated drought stress [115]. *OsMYB3* from rice was observed to enhance tolerance against cold, while *OsMYB2* was observed to enhance tolerance against dehydration, cold and salt stresses [116]. Expression of *OsMYB48-1* can be distinctly induced through PEG, ABA, H_2_O_2_ and dehydration, whereas only slight induction was observed after cold and high salinity stresses. However, overexpression of *OsMYB48-1* could enhance the drought and salinity tolerance in rice [117]. Similarly, wheat *TaMYB1D* was proven to be effective to enhance the tolerance against drought and oxidative stresses through regulating phenylpropanoid metabolism [118].

Among secondary metabolite, flavonoids and sinapate esters are key compounds in absorbing UV-B and enhance tolerance of plants from harmful UV-radiation effects in UV-B impaired mutant of *Arabidopsis* [119]. Initial targets of UV-B include lipids, proteins and nucleic acids and their damage is compromised by producing higher number of flavonoids and sinapate esters in plants under low exposure of UV-B light [119,120]. Many MYB TFs were observed to be regulated by mediating the phenylpropanoid metabolism to protect the plants against light and other stresses [120]. *AtMYB4*-mutated *Arabidopsis* plants showed enhanced tolerance to UV-B exposure because *MYB4* represses the function of gene producing cinnamate 4-hydroxylase enzyme, which is involved in biosynthesis of hydroxycinnamate ester, while another protein *AtMYB7* had a positive effect in producing phenylpropanoid compounds to absorb UV-B radiation [119]. Thus, it was concluded that both *AtMYB4* and *AtMYB7* are involved in regulating the balance for the production and accumulation of sunscreen compounds for radiation tolerance of *Arabidopsis* [119]. In addition, overexpression of *GmMYB12B2*, a soybean R2R3-MYB TF, rendered tolerance to UV radiation and salt stress in transgenic *Arabidopsis* [121].

### 4.3. Function and Expression Pattern of MYB TFs under Biotic Stresses

In addition to abiotic stress response, MYB TFs were also observed to be involved against biotic stresses. In *Arabidopsis*, *AtMYB102* was observed to enhance susceptibility against GPA (green-peach aphids) infestation [122]. *AtMYB96* acts as important molecular link between SA and ABA crosstalk, through which it can enhance the resistance against pathogens in *Arabidopsis* [123]. Beneficial microorganisms also trigger the defense responses of plant and in their response *MYB72* (root specific) act as signaling pathways convergence node in *Arabidopsis* [124]. Overexpression of *TaRIM1* increased the resistance against infestation of *Rhizoctonia cerealis* in transgenic wheat [125]. A MYB TF *y1* (yellow seed 1) of sorghum produced the 3-deoxyanthocyanidin phytoalexins against the attack of *Colletotrichum sublineolum* in maize [126]. Overexpression of *GmMYB12B2*, a soybean R2R3-MYB TF, rendered tolerance to UV radiation and salt stress in transgenic *Arabidopsis* [121]. Moreover, induction of MYB TFs against insect infestation was also observed in chrysanthemum, such as the overexpression of *CmMYB15* through lignin accumulation, which can reduce the aphid proliferation [127]. Similarly, another MYB TF, *MdMYB30*, could enhance disease resistance through the regulation of wax biosynthesis in apple [128]. *CaPHL8*, a novel MYB TF, was proven to be effective for increasing pepper plants immunity against *Ralstonia solanacerum* infection [129]. *VdMYB1*, a member of R2R3-MYB TF, was observed as a positive activator of the defense response by activating the expression of *stilbene synthase gene2* (*VdSTS2*) against infection stress caused by *Erysiphe necator* fungus of grapevine [130]. Overall, MYB TFs play a crucial role in enhancing the tolerance of plants against all type of stresses via biotic stresses and abiotic stresses.

## 5. AP2/ERF TFs in Response to Stresses

### 5.1. Classification and Diversification of AP2/ERF Gene Family

The AP2/ERF (APETALA2/Ethylene response element binding factors) superfamily genes are characterized by the presence of highly conserved DNA-binding domain with 60–-70 amino acids in each domain [131]. In addition, presence of alanine and aspartate at position 9 and 14, respectively, governs the binding of the *cis*-element [132]. However, based on number of domains (single or double) existing in genes, AP2/ERF can be categorized into five major sub-groups: AP2, RAV (Related to ABI3/VP1), DREB (Dehydration responsive element binding protein), ERF (Ethylene responsive factors) and others [132]. Until now, numerous AP2/ERF TFs were identified in plant species, including six AP2/ERF TFs in sugarcane and 30 AP2/ERF TFs in *Arabidopsis* (Table 1).

### 5.2. Function and Expression Pattern of AP2/ERF TFs under Abiotic Stresses

AP2/ERF TFs play a significant role in several abiotic stresses tolerance most probably through ET mediated signaling cascade. The identified AP2/ERF TFs, known to be involved in abiotic tolerance, belonged to *Arabidopsis* [132]. Overexpression of wheat *TaDREB3-A1* gene displayed the enhanced tolerance in transgenic *Arabidopsis* plants against heat, drought and salt stresses [133]. An observation revealed that *Arabidopsis* plants overexpressing *RAP2.1* showed an enhanced sensitivity to cold and drought stresses, as evidenced that *RAP2.1* acting as an active transcriptional repressor in defense responses to these stresses in *Arabidopsis* through the down-expression of the desiccation/Cold-regulated (RD/COR) genes [134]. Another study showed that overexpression of a *BnaERF-B3-hy15mu3* mutant gene from a modified AP2/ERF TF from *Brassica napus* enhances freeze tolerance in transgenic *Arabidopsis*, as resulted from this mutant gene encoded for a factor exhibiting more binding activity with the GCC box element than the wild-type gene [135]. Two Groups DREBs (I and II) from *Brassica napus* regulated synergistically the drought-responsive element (DRE)-mediated signaling pathway by trans-active and trans-inactive at different stages of cold stress in a competitive manner [136]. Expression pattern of ERF2 and ERF3 impart the acclimation response against cold stress and transgenic *Arabidopsis* lines, also exhibiting low electrolyte leakage with improved root architecture [137]. Similarly, higher measurements for antioxidants were observed, while lower MDA contents were observed due to the upregulation of cold responsive genes (*VaERF080* and *VaERF087*) in *Arabidopsis* [138]. A recent study also showed higher antioxidants, lower ROS, and improved cold tolerance by overexpression of *BpERF13* in birch [139]. *OsDREB1G* overexpressed in rice exhibited strong cold tolerance, but did not exhibit significant increase in drought or salt tolerance [140]. 

AP2/ERF TFs have been identified and characterized for waterlogging in many plant species. Sugarcane aerial roots developed under waterlogging stress, contribute to maintaining higher root activity and a higher ethylene concentration that increased the sensitivity of adventitious root-forming tissues and plays a principal role in aerenchyma formation [141]. Transcriptomic profiling of maize illustrated that 38 out of 184 AP2/ERF genes were remarkably induced by waterlogging stress [142]. The *Arabidopsis AtRAP2.12* is a key transcriptional regulator of the core anaerobic response controlled hydraulic conductivity of root 1 (HCR1), thereby regulating hydraulic conductivity (Lpr) and hypoxia responsive genes hence modulating the resilience of plants to multiple flooding scenarios [143]. Similarly, *ZmERFB180* [144] and *TaERFVII.1* [145] conferred waterlogging tolerance in maize and wheat, respectively. Three *ThRAP2.3/15/39* genes also played a vital role to cope with the waterlogging stress in the *Taxodium* hybrid [146].

The expression analysis in *Populus euphratica* depicted that upregulated expression of *PeDREBa* exhibited significantly higher values for physio-morphological traits and signal responsive regulation of drought and salt stress [147]. A soybean *GmDREB2* overexpressed in *Arabidopsis* alleviated the drought stress and salt toxicity through binding specifically to the *cis*-element [148]. Another soybean *GmDREB1* activated the expression of numerous soybean-specific stress-responsive genes i.e., an ABA receptor family protein (*GmPYL21)* and translation-related genes such as ribosomal proteins under diverse abiotic stress conditions [149]. Transcriptomic analysis revealed that *AP2si6* appeared to be more dynamically expressed in transgenic *Sesamum indicum* to cope with signal responsive water deficiency [150]. Overexpression of *OsDREP1* [151] and *AtDREB2A CA* [152] conferred drought endurance in rice and sugarcane, respectively. Transcriptional profiling of wheat plants has revealed that *Dreb2* genes were differentially upregulated in the presence of ABA responsive *cis*-element to cope with drought stress [153]. *TaDREB1* genes were induced as a result of DRE-binding protein (a transcript activator), thereby improving the tendency of wheat plants to tolerate the osmotic variations [154]. Transgenic wheat and barley carrying two DREB/CBF genes *TaDREB3* and *TaCBF5L* under the stress-responsive promoters *HDZI-3* and *HDZI-4* showed enhanced tolerance towards frost in wheat and drought resistance in barley [155]. Expression of *SodERF3* was induced by ABA, salt stress and wounding in sugarcane and overexpression of this gene in transgenic tobacco plants increased tolerance to drought and osmotic stresses [156]

### 5.3. Function and Expression Pattern of AP2/ERF TFs under Biotic Stresses

AP2/ERF TFs were associated with the regulation of disease resistance in plants. Overexpression of *TaPIE1* [157], *OsEREBP1* [158], *Soly106* [159], *GmERF113* [160] and *OsERF83* [161] genes, in response to JA, SA or ET mediated signal transduction pathways, were believed to be effective against pathogen infection. *HvRAF*, a novel AP2/ERF TF in barley, had dual regulatory functions under biotic (*R. solanacearum*) and abiotic (salinity) stresses [162]. In wheat, overexpression of *TaPIEP1* was significantly induced in response to *Bipolaris sorokiniana* infection. The transgenic plants with *TaPIEP1* overexpression were exhibited significantly higher resistance against fungal pathogen [163]. In addition to NAC and WRKY, several AP2/ERF TFs were differently expressed in resistant (CLN2777A) and susceptible (TMXA48-4-0) tomato cultivars in response to TYLCV [164]. A pepper *CaERF5* played a crucial role in protecting transgenic tobacco plants against *R. solanacearum* infection [165]. More recently, significantly higher resistance in *Arabidopsis* was observed by ectopic expression of an apple *MdERF11* in response to *Botryosphaeria dothidea* [166]. Collectively, AP2/ERF TFs play a vital role in abiotic and biotic stresses endurance through different stress-mediated signal transduction pathways. 

## 6. Conclusion and Future Prospects

TFs play pivotal roles at transcriptional level by either suppressing or activating genes under diverse stresses. Approximately 7% coding capacity of vascular plant genome is attributed to TFs for regulating genes at transcriptional level [167]. Thousands of TFs were identified in plants, while major TF families (WRKY, NAC, MYB, AP2/ERF, etc.), mediated through different signal transduction pathways, have been used to cope with abiotic and biotic stresses in various crops during the last two decades. However, more extensive field studies need to be conducted to identify the applications of TF genes in producing stress-resistant crops with high productivity to ensure food security. Although TFs as mediators of stresses have been used to produce stress-tolerant plants, advanced gene editing technologies such as CRISPR/Cas9 are powerful tools and can be explored in the future. To date, apart from the WRKY gene family [13], NAC, MYB and AP2/ERF TFs have not been identified in *Saccharum* species at a genome wide level due to its complex polyploid genome. There is an urgent need for employing molecular breeding tools to improve sugarcane cultivars, a vegetative propagation crop, with narrow genetic pools and a complicated genome. In addition, plant epigenetics, which is a conserved regulatory mechanism in gene expression, includes DNA methylation, histone modification, chromatin remodeling and noncoding RNA etc., which is an emerging tool to better understand the biological processes in sugarcane environmental responses.

## Figures and Tables

**Table 1 plants-09-00491-t001:** Gene numbers of WRYK, NAC, MYB and AP2/ERF transcription factors in land organisms ^a^.

Plant Species	Type of Organism	WRKY	NAC	MYB	AP2/ERF
*Saccharum* spp. (Sugarcane)	Monocot plant	39	44	36	6
*Sorghum bicolor* (Sorghum)	Monocot plant	134	180	145	42
*Triticum aestivum* (Wheat)	Monocot plant	171	263	263	43
*Oryza sativa* subsp. *japonica* (Rice)	Monocot plant	128	170	130	22
*Hordeum vulgare* (Barley)	Monocot plant	126	150	99	34
*Zea mays* (Maize)	Monocot plant	161	189	203	54
*Zoysia matrella* (Manila grass)	Monocot plant	269	313	293	49
*Brachypodium distachyon* (False brome)	Monocot plant	134	186	163	64
*Beta vulgaris* (Sugar beet)	Dicot Plant	49	59	68	14
*Arabidopsis thaliana*	Dicot Plant	90	138	168	30
*Nicotiana tabacum* (Tobacco)	Dicot Plant	210	280	319	93
*Jatropha curcas* (Physic nut	Dicot Plant	61	97	115	19
*Cucumis sativus* (Cucumber)	Dicot Plant	88	102	134	27
*Solanum lycopersicum* (Tomato)	Dicot Plant	81	101	140	27
*Vitis Vinifers* (Grape)	Dicot Plant	59	71	138	19
*Gossypium hirsutum* (Cotton)	Dicot Plant	238	306	441	59
*Glycine max* (Soybean)	Dicot Plant	296	269	430	99
*Sesamum indicum* (Sesame)	Dicot Plant	88	105	168	43
*Camelina sativa* (Flax)	Dicot Plant	224	350	402	93
*Brassica napus* (Rapeseed)	Dicot Plant	285	411	489	57
*Chlamydomonas reinhardtii*	Single-celled green alga	2	0	16	12
*Gonium pectoral*	Single-celled green alga	1	0	12	3
*Ostreococcus lucimarinus*	Single-celled green alga	2	0	11	3
*Selaginella moellendorffii*	Moss	19	22	24	10
*Physcomitrella patens*	Moss	117	142	180	44

^a^ The data was summarized from plant transcription factor database, 2020.

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
