# Peer review of "Transcription Factors in Plant Stress Responses: Challenges and Potential for Sugarcane Improvement"

_plants, 2020, doi:10.3390/plants9040491_

Round 1
Reviewer 1 Report
This is a fairly interesting review that is fairly well written, although it could be improved with some English editing. Overall, I thought the text was acceptable, but I have some concerns with the table and figure.
Table 1: For each family (e.g. WRKY), the number of genes present in a given organism and their expression is listed. For example, wheat has 171 WRKY genes expressed in stem and leaves. Are all 171 genes expressed in stem and leaves? This seems very unlikely. Some of the 171 genes may be pseudogenes and not expressed. They are also likely to be differentially expressed. What does expressed in "stem and leaves" mean then? The authors should explain how they decided how things are listed in the expression column. For example, does "stem and leaves" mean that >75% of the WRKY genes in wheat expressed in only stem and leaves? What is the cutoff value?
Figure 1: I do not see the point of this figure and it seems misleading. First, the legend says "The number of AP2/ERF TFs in sugarcane and other crops", but a pie graph is shown that gives proportions, not numbers. Numbers would be much more informative (although they are given in parentheses) and would be better visualized with bar graphs and not a pie chart. Also, 90% of the TFs are from "other" crops. What are these other crops? What is the point of including "other" in the figure? Usually if something is listed as "other" in a pie chart it is a very small proportion (<10%), not 90%. All the species in "other" should be given or it should be left out all together.
Author Response
Comment 1: This is a fairly interesting review that is fairly well written, although it could be improved with some English editing. Overall, I thought the text was acceptable, but I have some concerns with the table and figure. Table 1: For each family (e.g. WRKY), the number of genes present in a given organism and their expression is listed. For example, wheat has 171 WRKY genes expressed in stem and leaves. Are all 171 genes expressed in stem and leaves? This seems very unlikely. Some of the 171 genes may be pseudogenes and not expressed. They are also likely to be differentially expressed. What does expressed in "stem and leaves" mean then? The authors should explain how they decided how things are listed in the expression column. For example, does "stem and leaves" mean that >75% of the WRKY genes in wheat expressed in only stem and leaves? What is the cutoff value? Response: We removed the column having the information of genes expression and added the TFs information from four important crop plants. Comment 2: Figure 1: I do not see the point of this figure and it seems misleading. First, the legend says "The number of AP2/ERF TFs in sugarcane and other crops", but a pie graph is shown that gives proportions, not numbers. Numbers would be much more informative (although they are given in parentheses) and would be better visualized with bar graphs and not a pie chart. Also, 90% of the TFs are from "other" crops. What are these other crops? What is the point of including "other" in the figure? Usually if something is listed as "other" in a pie chart it is a very small proportion (Reviewer 2 Report
This review on roles of transcription factors in grasses under abiotic and biotic stress conditions is well summarized. However, there are some editing mistakes in the text. Please carefully edit the text again.
Author Response
Comment: This review on roles of transcription factors in grasses under abiotic and biotic stress conditions is well summarized. However, there are some editing mistakes in the text. Please carefully edit the text again. Response: We have tried our best to correct the mistakes in text.Reviewer 3 Report
The manuscript provides a review of plant transcription factors (TFs) with a comprehensive summary of responses in the model organism Arabidopsis. The manuscript describes how TFs have been less explored in sugarcane and argues that filling this gap can be used in crop improvement programs in response to abiotic and biotic stress.
The manuscript provides no independent research results, however, argues that more research into TFs for sugarcane should be made.
I am not qualified to evaluate if this review provides a sufficient novel product to merit publication. However, the conclusion that more research into FTs for sugarcane is needed, might perhaps be told without the need for quite so many examples of FTs in Arabidopsis?
Without myself being a native English speaker, I notice that the manuscript needs a major review of correct English grammar. Plural forms and third-person forms of verbs need a careful review. Besides the grammatical errors, the English text is understandable and easy to read.
Line 57, involving -> involved
Line 120, Overexpressing of VaWRKY45 that was localized in the nucleus, inhibited ...
Line 136, deficit -> deficiency
... etc
Check the reference list syntax. For reference 5 and 31, the year is not formatted in bold. Recommend to include the DOI for the cited references.
Author Response
Comment 1: The manuscript provides a review of plant transcription factors (TFs) with a comprehensive summary of responses in the model organism Arabidopsis. The manuscript describes how TFs have been less explored in sugarcane and argues that filling this gap can be used in crop improvement programs in response to abiotic and biotic stress. The manuscript provides no independent research results, however, argues that more research into TFs for sugarcane should be made. I am not qualified to evaluate if this review provides a sufficient novel product to merit publication. However, the conclusion that more research into FTs for sugarcane is needed, might perhaps be told without the need for quite so many examples of FTs in Arabidopsis? Response: We added more recent findings from sugarcane and other field crops, despite the expression pattern of various TFs in response to abiotic and biotic stresses in model and non-model plant species have been incorporated in the manuscript. Our manuscript described the importance of various TFs against abiotic and biotic stresses in different plant species and this can be helpful for sugarcane improvement programs. Comment 2: Without myself being a native English speaker, I notice that the manuscript needs a major review of correct English grammar. Plural forms and third-person forms of verbs need a careful review. Besides the grammatical errors, the English text is understandable and easy to read. Response: Grammatical errors from the manuscript have been corrected after confirmation from the native English speaker. Comment 3: Line 57, involving -> involved Response: Corrected as you required. Comment 4: Line 120, Overexpressing of VaWRKY45 that was localized in the nucleus, inhibited ... Response: Corrected as you required. Comment 5: Line 136, deficit -> deficiency Response: Corrected as you required. Comment 6: Check the reference list syntax. For reference 5 and 31, the year is not formatted in bold. Recommend to include the DOI for the cited references. Response: References have been revised accordingly. We did not include the DOI for the cited references based on the instructions from this journal.Reviewer 4 Report
Dear authors,
This manuscript is a very good revision that provides an overview and the state-of-the-art in what concerns the role of the TFs in plants under several stresses. In fact, the WRKY, NAC, MYB, and AP2/ERF TF families are very important in all plant gene expression regulation under stress responses. At this stage, the overall manuscript presentation convinces me to be published as it stands.
Author Response
Comment: This manuscript is a very good revision that provides an overview and the state-of-the-art in what concerns the role of the TFs in plants under several stresses. In fact, the WRKY, NAC, MYB, and AP2/ERF TF families are very important in all plant gene expression regulation under stress responses. At this stage, the overall manuscript presentation convinces me to be published as it stands. Response: Noted with thanks.Round 2
Reviewer 3 Report
Thanks for the English syntax editing. The manuscript looks much better now (to a still non-native speaker). However, I still find a few English syntax issues the authors might want to look at:
Line 157: “GhWRKY44, ..., provide[s] enhanced” --- [it provides]
Line 186: “While in atypical NAC TFs, [the] C-terminal region[s] contain[s]” --- “[the] region contains” or “[the] region[s] contain”.
Line 204: Various members of NAC play [a] significant role --- or “play significant roles”
Line 281: “Many of MYB TFs have been observed to play [a] significant role” --- or “to play significant role[s]”
Line 355: “, and [Oïƒ o]thers” --- no need to capitalize “others” mid-sentence
Author Response
We corrected these errors as you required. Thank you so much.